# Localizing Active Objects from Egocentric Vision with Symbolic World Knowledge

**Te-Lin Wu*   Yu Zhou*   Nanyun Peng**
University of California, Los Angeles
{telinwu,violetpeng}@cs.ucla.edu, yu.zhou@ucla.edu

## Abstract

The ability to actively ground task instructions from an egocentric view is crucial for AI agents to accomplish tasks or assist humans. One important step towards this goal is to localize and track *key active objects* that undergo major state change as a consequence of human actions/interactions in the environment (*e.g.*, localizing and tracking the 'sponge' in video from the instruction *"Dip the sponge into the bucket."*) without being told exactly what/where to ground. While existing works approach this problem from a pure vision perspective, we investigate to which extent the language modality (*i.e.*, task instructions) and their interaction with visual modality can be beneficial. Specifically, we propose to improve phrase grounding models' (Li* et al., 2022) ability in localizing the active objects by: (1) learning the role of *objects undergoing change* and accurately extracting them from the instructions, (2) leveraging pre- and post-conditions of the objects during actions, and (3) recognizing the objects more robustly with descriptional knowledge. We leverage large language models (LLMs) to extract the aforementioned action-object knowledge, and design a per-object aggregation masking technique to effectively perform joint inference on object phrases with symbolic knowledge. We evaluate our framework on Ego4D (Grauman et al., 2022) and Epic-Kitchens (Dunnhofer et al., 2022) datasets. Extensive experiments demonstrate the effectiveness of our proposed framework, which leads to > 54% improvements in all standard metrics on the TREK-150-OPE-Det localization + tracking task, > 7% improvements in all standard metrics on the TREK-150-OPE tracking task, and > 3% improvements in average precision (AP) on the Ego4D SCOD task.

## 1 Introduction

Recent technological advancements in smart glasses (and headsets) from industry leaders such

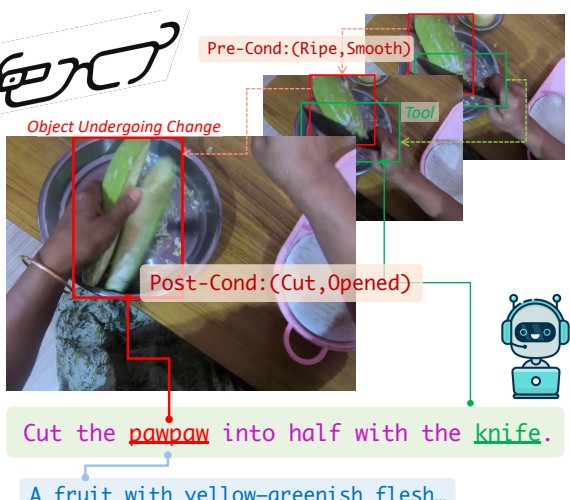

Figure 1: **Active object grounding** is the task of localizing the active objects undergoing state change (OUC). In this example action instruction "cut the pawpaw into half with the knife", the AI assistant is required to firstly infer the OUC (pawpaw) and the Tool (knife) from the instruction, and then localize them in the egocentric visual scenes throughout the action trajectories. Symbolic knowledge including pre/post conditions and object descriptions can bring additional information to facilitate the grounding.

as Meta, Google, and Apple have attracted growing research in on-device AI that can provide *just-in-time* assistance to human wearers. [1] While giving (or receiving) instructions during task execution, the AI assistant should *co-observe* its wearer's first-person (egocentric) viewpoint to comprehend the visual scenes and provide appropriate assistance. To accomplish this, it is crucial for AI to first be able to localize and track the objects that are undergoing significant state change according to the instruction and/or actions performed. For example in Figure 1, it can be inferred from the instruction that the object undergoing change which should be **actively grounded** and **tracked** is the *pawpaw*.

Existing works have focused on the visual modality alone for such state change object localization tasks, including recognizing hand-object interac-

---

*The authors contribute equally, alphabetical order.

[1]Code at: https://github.com/PlusLabNLP/ENVISION

tions (Shan et al., 2020a) and object visual state changes (Alayrac et al., 2017). However, it remains under-explored whether the visual modality by itself is sufficient for providing signals to enable robust state change object localizing/tracking without enhanced signals from the textual modality. While utilizing a phrase grounding model (Liu et al., 2022, 2023) is presumably a straightforward alternative, it leaves unanswered questions of which mentioned objects/entities in the instruction are supposedly the one(s) that undergo major state changes, *e.g.*, the *pawpaw* in Figure 1 instead of the *knife* is the correct target-object. Furthermore, how visual appearances of the objects can help such multimodal grounding is yet to be investigated.

In light of this, we tackle the active object grounding task by first extracting target object mentionsfrom the instructions using large language models (ChatGPT (OpenAI, 2023)) with a specifically designed prompting pipeline, and then finetuning an open-vocabulary detection model, GLIP (Li* et al., 2022), for visual grounding.

We further hypothesize that additional action- and object-level symbolic knowledge could be helpful. As shown in Figure 1, state conditions *prior to* (*pre-conditions*: which indicate pre-action states) and *after* (*post-conditions*: which suggest at past state changes) the execution of the action are often considered when locating the objects, especially when the state changes are more visually significant. Furthermore, generic object knowledge including visual descriptions (*e.g.*, *"yellow-greenish flesh"*), are helpful for uncommon objects.[2] Based on this hypothesis, we prompt the LLM to obtain pre- and post-conditions on the extracted object mentions, along with a brief description focusing on specific object attributes.

To improve the grounding models by effectively using all the aforementioned action-object knowledge, we design an object-dependent mask to separately attribute the symbolic knowledge to its corresponding object mentions for training. During inference time, a pre-/post-condition dependent scoring mechanism is devised to aggregate the object and the corresponding knowledge logit scores to produce a joint inference prediction.

We evaluate our proposed framework on two

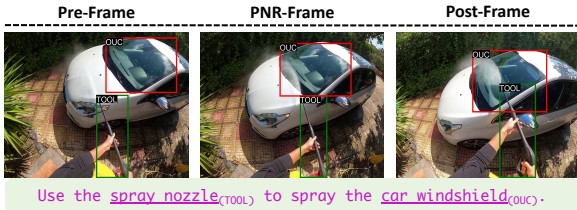

Figure 2: **Ego4D SCOD active grounding:** Example object undergo change (OUC) due to the instructed actions and associated Tools, spanning: the pre-condition, point-of-no-return (PNR) and post-condition frames.

narrated egocentric video datasets, Ego4D (Grauman et al., 2022) and Epic-Kitchens (Damen et al., 2022) and demonstrate strong gains. Our main contributions are two folds: (1) We design a sophisticated prompting pipeline to extract useful symbolic knowledge for objects undergoing state change during an action from instructions. (2) We propose a joint inference framework with a per-object knowledge aggregation technique to effectively utilize the extracted knowledge for improving multimodal grounding models.

## 2 Tasks and Terminologies

**Active Object Grounding.** For both robotics and assistant in virtual or augmented reality, the AI observes (or co-observes with the device wearer) the visual scene in the *first-person* (*egocentric*) point of view, while receiving (or giving) the task instructions of what actions to be performed next. To understand the context of the instructions as well as engage in assisting the task performer's actions, it is crucial to closely follow the key objects/entities that are involved in the actions undergoing major state change.[3] We term these actively involved objects as **objects undergoing change (OUC)**, and what facilitate such state change as **Tools**.

**Tasks.** As there is not yet an existing resource that directly studies such active instruction grounding problem in real-world task-performing situations, we *re-purpose* two existing egocentric video datasets that can be seamlessly transformed into such a setting: **Ego4D** (Grauman et al., 2022) and **Epic-Kitchens** (Damen et al., 2018). Both come with per-time-interval annotated narrations transcribing the main actions occurred in the videos.[4]

**Ego4D: SCOD.** According to Ego4D's definition,

---

[2]This should not contradict with the application where an assistive AI judges if the outcomes of the actions are desirable, as here we are only using general commonsensical conditions generated by an LLM, while in reality there can be more subtle and task-dependent conditions that need to be examined.

[3]State change can come from objects' physical properties such as *composition*, *textures*, and *functionalities*; as well as attributes such as *sizes*, *shapes*, and *physical affordances*.

[4]The narrations are paraphrased as imperative instructions.

object state change can encapsulate both spatial and temporal aspects. There is a timestamp that the state change caused by certain actions start to occur, *i.e.*, the **point-of-no-return (PNR)**. Ego4D's **state change object detection (SCOD)** subtask then defines, chronologically, three types of frames: the **pre-condition (Pre)**, the **PNR**, and the **post-condition (Post)** frames, during a performed action. Pre-frames capture the prior (visual) states where a particular action is allowed to take place, while post-frames depict the outcomes caused by the action, and hence also record the associated object state change. Each frame annotated with its corresponding frame-type is further annotated with bounding boxes of the OUC (and Tools, if applicable), that is required to be regressed by the models. Figure 2 shows an exemplar SCOD data point.

Our re-purposed active grounding task is thus as follows: *Given an instructed action and one of a Pre/PNR/Post-typed frames, **localize (ground)** both the OUC(s) and Tool(s) in the visuals.* While the official SCOD challenge only evaluates the PNR frame predictions, we consider all (Pre, PNR, and Post) frames for both training and inference.

**Epic-Kitchens: TREK-150.** TREK-150 object tracking challenge (Dunnhofer et al., 2022, 2021) enriches a subset of 150 videos from the Epic-Kitchens (Damen et al., 2018, 2022) dataset, with densely annotated per-frame bounding boxes for tracking a *target object*. Since the Epic-Kitchens also comprises egocentric videos capturing human performing (specifically kitchen) tasks, the target objects to track are exactly the OUCs per the terminology defined above. *Hence, given an instructed action, the model is required to **ground and track the OUC in the egocentric visual scenes.**[5]*

It is worth noting that some OUCs may occasionally go "in-and-out" of the egocentric point of view (PoV), resulting in partial occlusion and/or full occlusion frames where no ground truth annotations for the OUCs are provided. Such frames are excluded from the final evaluation. And in Section 4.2.3 we will show that our proposed model is very successful in predicting the objects when they come back due to the robustness of our symbolic joint inference grounding mechanism.

---

[5] Unlike Ego4D SCOD task, TREK-150 does not contain any defined Pre/PNR/Post frames. Our proposed model is trained to perform joint inference and autonomously decide which of the pre- and post-conditions to weigh more based on the frame image and instructed action. And hence, in the TREK-150 task, frame-type information is not required.

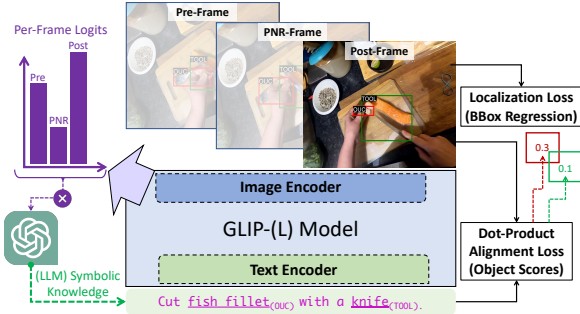

Figure 3: **Overview of proposed framework** that comprises a base multimodal phrase grounding model (GLIP), a frame-type predictor, a knowledge extractor leveraging LLMs (GPT), and predictions supervised by both bounding box regression of the objects and their ranked scores.

## 3 Method

Figure 3 overviews the proposed framework, consisted of: (1) A base **multimodal grounding architecture**, where we adopt a strong *open vocabulary object detection module*, GLIP (Li* et al., 2022). (2) A **frame-type prediction** subcomponent which adds output projection layers on top of GLIP to utilize both image (frame) and text features to predict of which frame-type (Pre/PNR/Post) is currently observed. (Section 3.2.1) (3) A **prompting pipeline** that is engineered to extract useful action-object knowledge from an **LLM (GPT)**. (Section 3.2) (4) A **per-object knowledge aggregation** technique is applied to GLIP's word-region alignment contrastive training. (Section 3.3)

### 3.1 Adapting GLIP

GLIP (Li* et al., 2022; Zhang* et al., 2022) achieves open vocabulary object detection by pre-training on a contrastive phrase grounding objective. Specifically, GLIP extends the text(caption)-to-image dot product matching objective from CLIP (Radford et al., 2021) to a **word-region**-level alignment objective. For some (tokenized) words of the textual description of an image, there are certain image region(s) that could be grounded to, while other regions are viewed as the negative samples for the CLIP-like alignment contrastive learning. During pretraining, GLIP utilizes both phrase grounding datasets (Ordonez et al., 2011; Plummer et al., 2015; Sharma et al., 2018) and object detection datasets (Krishna et al., 2017; Krasin et al., 2017; Shao et al., 2019).[6]

**Contrastive Learning.** We illustrate the GLIP

---

[6] The descriptions of object detection are a simple concatenation of all the available object class labels.

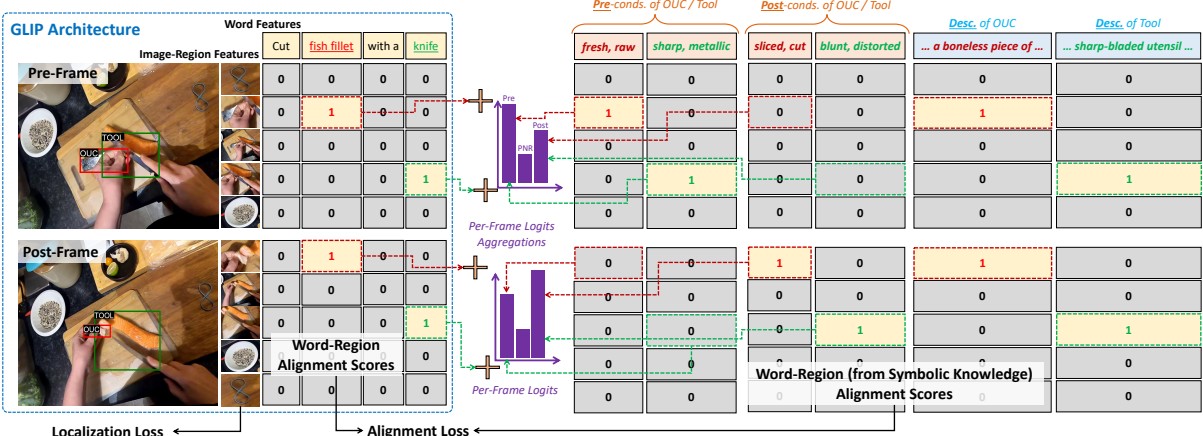

Figure 4: **Model architecture (knowledge-enhanced grounding):** On the left depicts the word-region alignment (contrastive) learning of the base GLIP architecture, where the model is trained to align the encoded latent word and image features with their dot-product logits being supervised by the positive and negative word-region pairs. On the right illustrates the enhanced object-knowledge grounding. During training we apply an object-type dependent mask to propagate the positive alignment supervisions; while during inference time the frame-type predictor (offline trained by the encoded textual and image features) acts as a combinator to fuse dot product-logit scores from both (extracted) object phrases and corresponding knowledge. (Note that for simplicity we do not fully split some phrases into individual words.)

training adapted to our task in Figure 4 left half. Notice that for simplicity we do not fully expand the tokenized word blocks, *e.g.*, *"fish fillet"* should span two words where each word ( *fish"* and *"fil-let"*) and its corresponding region is all regarded as the positive matching samples. The model is trained to align the encoded latent word and image features[7] with their dot-product logits being supervised by the positive and negative word-region pairs. The alignment scores will then be used to score (and rank) the regressed bounding boxes produced by the image features, and each box will feature an object-class prediction score. Concretely, for $j$th regressed box, its grounding score to a phrase $W = \{w\}_{1:T}$ is a mean pooling of the dot-products between the $j$th region feature and all the word features that compose such a phrase: $\mathbf{S}_j^{box} = \frac{1}{T} \sum_i^T \mathbf{I}_j \cdot \mathbf{W}_i$. In this work, we mainly focus on the OUC and Tool object classes, *i.e.*, each textually-grounded region will further predict whether it is an OUC or Tool class.

## 3.2 LLM for Action-Object Knowledge

**Pipeline.** As illustrated in Figure 5, we implement an LLM query pipeline to extract active entities and relevant symbolic knowledge from an instructional caption. To account for GPT's verbose tendency, we forcibly instruct GPT to produce the active objects (OUC and/or Tool) following a specifically designed format and then apply heuristic-based

post-processing to further refine the extractions.[8] Conditioned on the extracted OUC (and Tool), two additional queries are made to generate: (1) the symbolic pre- and post-conditions of such objects induced by the actions, and (2) brief descriptions characterizing the objects and their attributes. Interestingly, we empirically find it beneficial to situate GPT with a role, *e.g.*, *"From the first-person view."*

**GPT Intrinsic Evaluation.** In Table 2, we automatically evaluate the OUC/Tool extraction of GPT against the labelled ground truth entities in both datasets. We report both exact (string) match and word overlapping ratio (as GPT often extracts complete clauses of entities), to quantify the robustness of our GPT active entity extractions.

Table 3 reports human evaluation results of GPT symbolic knowledge, including pre-/post-conditions and descriptions. Evaluation is based on two binary metrics, namely: (1). Textual Correctness: *"Based on text alone, does the knowledge make sense?"* and (2). Visual Correctness: *"Does the conds./desc. match the image?"* Despite impressive performance on both intrinsic evaluations, we qualitatively analyze in Table 1 some representative cases where GPT mismatches with annotations or humans, including cases where GPT's answer is actually more reasonable than the annotations.

### 3.2.1 Incorporating Knowledge

**Adding Knowledge.** We use the following schema to enrich the instruction with the obtained knowl-

---

[7]For details of GLIP's multimodal fusion technique, we refer the readers to Li* et al. (2022); Zhang* et al. (2022).

[8]More details are in Append. Sec. A.2.

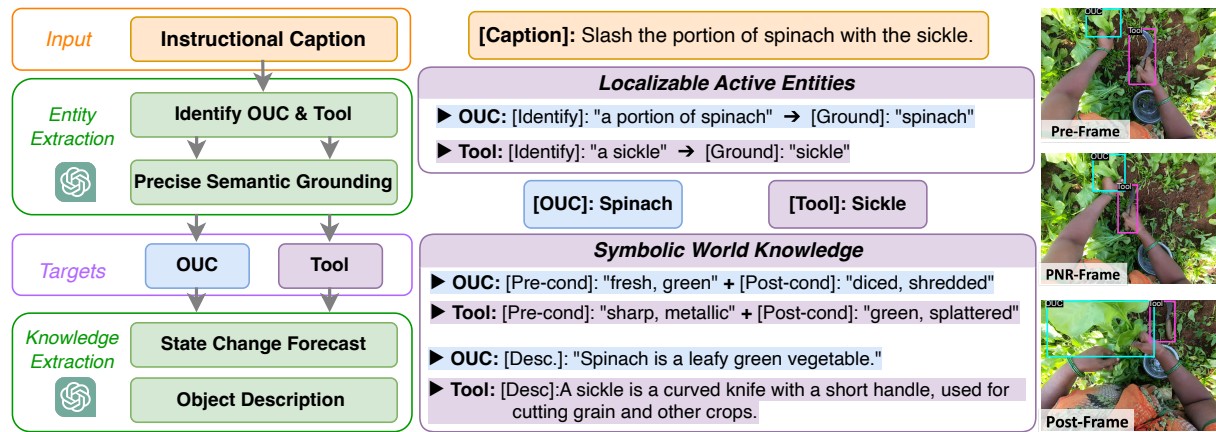

Figure 5: **The GPT knowledge extraction pipeline.** Demonstrated through an example from the Ego4D SCOD Dataset.

| Examples | Explanations |
|---|---|
| **GPT**: Pick up some green papers(OUC) from the table. [No Tool] 
 **Desc.**: "*Green papers* are consultation documents issued by government." | Without visual knowledge input, GPT is not robust to phrase ambiguity, leading to undesirable definition. |
| **GPT**: Cut the fish fillet(OUC) with a knife(TOOL) 
 **OUC**: [Pre-cond]: "fresh, raw" [Post-cond]: "sliced, cut" 
 **Tool**: [Pre-cond]: "Sharp, metallic" [Post-cond]: "Blunt, distorted" | LLMs may hallucinate exaggerated state changes, in this case claiming the knife to be "blunt, distorted" after a single use, which is unreasonable. |
| **GPT**: Hold the iron(OUC) on the ironing board with your hand. [No Tool] 
 **Ego4D gt-label**: [OUC]: "pants" [Tool]: "iron" | "Pants" is not mentioned in the narration, GPT fails to capture OUC due to text narration reporting bias. |
| **GPT**: Spin the mop(OUC) in the mop bucket spinner(Tool). 
 **Ego4D gt-label**: [OUC]: "mop" [Tool]: "mop" | GPT prediction is more reasonable compared to the Ego4D ground truth label. |

Table 1: **Qualitative Analysis of GPT Knowledge Extraction:** Examples of cases where GPT-extracted symbolic knowledge are wrong or conflict with Ego4D annotations. Here the GPT-extracted or dataset-annotated knowledge are displayed in GREEN if they match human analysis and RED otherwise. Explanations for each example are provided on the right.

| Object Type | Ego4D SCOD | | TREK-150 | |
|---|---|---|---|---|
| | EM (%) | Overlap. | EM (%) | Overlap. |
| OUC | 77.8 | 88.6 | 76.0 | 94.3 |
| Tool | 60.3 | 88.5 | — | — |

Table 2: **Automatic evaluation of GPT entity extraction.** Abbreviations: **EM**: exact string matching; **Overlap:** The ratio of GPT extractions fully covering the ground truth phrases

| Knowledge Type | Ego4D SCOD | | TREK-150 | |
|---|---|---|---|---|
| | Textual | Visual | Textual | Visual |
| Pre-Cond. | 86.5 | 81.6 | 83.0 | 79.9 |
| Post-Cond. | 75.2 | 70.3 | 76.6 | 73.5 |
| Desc. | 98.9 | 91.4 | 99.2 | 95.3 |

Table 3: **Human evaluation of GPT symbolic knowledge extraction.** Abbreviations: **Textual**: i.e. "textual correctness" "Based on text alone, does the GPT conds./desc. make sense?"; **Visual**: i.e. "visual correctness": "Does the GPT conds./desc. match what is shown in the image?"

edge: "*{instr.}* [SEP] object/tool (pre/post)-state is *{conds.}* [SEP] object/tool description is *{desc.}*", where [SEP] is the separation special token; *{conds.}* and *{desc.}* are the pre-/post-condition and object definition knowledge to be filled-in. Empirically, we find diffusing the post-condition knowl-

edge to PNR frame yield better results. As Figure 4 illustrates (omitting some prefixes for simplicity), we propagate the positive matching labels to object/tool's corresponding knowledge. In the same training mini-batch, we encourage the contrastiveness to focus on more detailed visual appearance changes grounded to the symbolic condition statements and/or descriptions, by sampling frames from the same video clips with higher probability.

**Frame-Type Prediction.** Using both the encoded textual and image features, we learn an additional layer to predict the types of frames conditioned on the associated language instruction. Note that the frame-type definition proposed in Ego4D should be generalizable outside of the specific task, *i.e.*, these frame types could be defined on any kinds of action videos. In addition to the annotated frames in SCOD, we randomly sub-sample nearby frames within 0.2 seconds (roughly 5-6 frames) to expand the training data. The frame-type prediction achieves a 64.38% accuracy on our SCOD test-set, which is then directly applied to the TREK-150 task for deciding the amount of pre- and post-condition

knowledge to use given the multimodal inputs.

### 3.3 Object-Centric Joint Inference

**Masking.** As illustrated in Figure 4, a straightforward way to assign symbolic knowledge to its corresponding object type respectively is to construct a **per-object-type mask**. For example, an OUC mask $\mathbf{M}_{OUC}$ will have 1s spanning the positions of the words from condition (*e.g.*, *"fresh,raw"* of the OUC *"fish fillet"* in Figure 4) and descriptive knowledge, and 0s everywhere else. We omit the knowledge prefixes in Section 3.2.1 (*e.g.*, the phrase *"object state is"*) so that the models can concentrate on grounding the meaningful words. Such mask for each object type can be deterministically constructed to serve as additional word-region alignment supervisions, and can generalize to object types outside of OUC and Tool (beyond the scope of this work) as the GPT extraction can clearly indicate the object-to-knowledge correspondences. In other words, we enrich the GLIP's phrase grounding to additionally consider symbolic knowledge during the contrastive training. Note that the mask is frame-type dependent, *e.g.*, $\mathbf{M}_{OUC}^{Pre}$ and $\mathbf{M}_{OUC}^{Post}$ will focus on their corresponding conditional knowledge.

**Aggregation.** During the inference time, we combine the frame-type prediction scores $\mathbf{S}^{fr}$ with the per-object mask to aggregate the dot-product logit scores for ranking the regressed boxes. Specifically, we have $\mathbf{S}_{OUC}^{box} = \sum_{fr} \mathbf{S}^{fr} * \mathbf{M}_{OUC}^{fr}$, where $\mathbf{S}$ is a 3-way logit and $fr \in \{\text{Pre}, \text{PNR}, \text{Post}\}$.

## 4 Experiments and Analysis

We adopt the **GLIP-L** variant (and its pretrained weights) for all of our experiments, where its visual encoder uses the Swin-L transformer (Liu et al., 2021). We train the GLIP-L with our framework primarily on the SCOD dataset, and perform a zero-shot transfer to the TREK-150 task.

### 4.1 Ego4D SCOD

#### 4.1.1 Experimental Setups

**Data Splits.** We split the official SCOD train set following a 90-10 train-validation ratio and use the official validation set as our primary test set.[9]

**Evaluation Metrics.** Following the original SCOD task's main settings, we adopt average precision

(AP) as the main evaluation metric, and utilize the COCO API (Lin et al., 2014) for metric computation. Specifically, we report AP, AP50, (AP at IOU$\geq$ 0.5) and AP75 (AP at IOU$\geq$ 0.75).

#### 4.1.2 Baselines

We evaluate three categories of baselines: (1) **Pure object detection** models, where the language instructions are not utilized. (2) **(Pseudo) referential grounding**, where certain linguistic heuristics are used to propose the key OUCs. (3) **GPT** with **symbolic knowledge**, where GPT is used to extract both the OUCs and Tools, with additional symbolic knowledge available to utilize.

**Pure Object Detection (OD).** We finetune the state-of-the-art model of the SCOD task from Chen et al. (2022) (**VidIntern**) on all types of frames (Pre, PNR, and Post) to serve as the pure object detection model baseline, which learns to localize the OUC from a strong hand-object-interaction prior in the training distribution. We also train an OD version of GLIP providing a generic instruction, *"Find the object of change."*, to quantify its ability to fit the training distribution of plausible OUCs.

**Pseudo Grounding (GT/SRL).** We experiment four types of models utilizing the instructions and certain linguistic patterns as heuristics: (1) We extract all the nouns using Spacy NLP tool (Honnibal and Montani, 2017) and randomly assign OUC to one of which (**Random Entity**). (2) A simple yet strong baseline is to ground the full sentence of the instruction if the only object class to be predicted is the OUC type (**Full-Instr.**). (3) Following (2), we hypothesize that the first argument type (ARG1) of the semantic-role-labelling (SRL) parses (Shi and Lin, 2019; Gardner et al., 2017) of most simple instructions is likely regarded as the OUC ((**SRL-ARG1**)). (4) Lastly, to quantify a possible upper bound of simple grounding methods, we utilize the annotated ground truth object class labels from SCOD task and perform a simple pattern matching to extract the OUCs and Tools. For those ground truth words are not easily matched, we adopt the ARG1 method from (3) (**GT-SRL-ARG1**).

**GPT-based.** For our main methods leveraging LLMs (GPT) and its generated action-object symbolic knowledge, we consider four types of combinations: (1) **GPT** with its extracted OUCs and Tools. (2) The model from (1) with additional utilization of object definitions (**GPT+Desc.**). (3) Similar to (2) but condition on generated pre- and

---

[9]The official test-set only concerns the PNR frame, and deliberately excluded narrations to make a vision only localization task, which is not exactly suitable for our framework.

| Base | Type | Method | Objects | Pre-Frame↑ | | | PNR-Frame↑ | | | Post-Frame↑ | | |
|---|---|---|---|---|---|---|---|---|---|---|---|---|
| | | | | AP | AP50 | AP75 | AP | AP50 | AP75 | AP | AP50 | AP75 |
| **VidIntern** | **OD** | — | OUC | 32.73 | 49.17 | 34.05 | 37.49 | 57.04 | 38.59 | 29.68 | 44.43 | 30.94 |
| | | | Tool | 16.39 | 23.43 | 17.25 | 16.53 | 24.51 | 17.14 | 14.03 | 21.70 | 14.44 |
| **GLIP-L** | **OD** | — | OUC | 26.91 | 42.83 | 27.86 | 29.74 | 47.70 | 30.47 | 24.13 | 38.74 | 24.71 |
| | **Instr.** | Zero-Shot on GTs | OUC | 20.18 | 32.97 | 20.63 | 19.51 | 32.34 | 19.39 | 19.34 | 31.07 | 19.88 |
| | | Random Entity | OUC | 25.90 | 42.17 | 26.20 | 26.85 | 44.21 | 26.80 | 24.45 | 39.17 | 25.10 |
| | | Full-Instr. | OUC | 32.45 | 51.62 | 33.34 | 33.78 | 54.44 | 34.49 | 31.30 | 49.23 | 32.42 |
| | | SRL-ARG1 | OUC | 36.41 | 54.93 | 37.65 | 38.32 | 58.07 | 39.41 | 33.59 | 49.99 | 34.90 |
| | | GT-SRL-ARG1 | OUC | 37.87 | 56.35 | 39.55 | 39.64 | 59.41 | 40.73 | 34.97 | 51.34 | 36.69 |
| | | | Tool | **45.53** | **71.22** | **46.27** | **43.70** | **68.96** | 44.54 | **43.76** | **69.56** | **44.04** |
| | | GPT | OUC | 37.46 | 56.05 | 38.96 | 39.07 | 59.17 | 40.13 | 34.77 | 51.34 | 36.35 |
| | | | Tool | 38.41 | 60.66 | 39.33 | 37.64 | 60.26 | 39.29 | 37.67 | 59.73 | 38.24 |
| | | GPT+Desc. | OUC | 36.97 | 56.16 | 38.35 | 38.49 | 59.38 | 39.41 | 34.09 | 51.18 | 35.56 |
| | | | Tool | 42.26 | 64.37 | 44.59 | 41.30 | 64.46 | 43.53 | 40.20 | 63.92 | 41.60 |
| | | GPT+Conds. | OUC | **38.65** | 57.55 | **40.16** | **40.19** | 60.39 | **41.56** | **35.40** | 52.15 | **37.11** |
| | | | Tool | 43.48 | 65.78 | 45.58 | 42.37 | 64.97 | 44.77 | 41.08 | 63.26 | 42.07 |
| | | w/o obj.-mask | OUC | 37.59 | 56.28 | 39.19 | 39.09 | 59.31 | 40.58 | 33.93 | 50.80 | 35.38 |
| | | GPT+Conds.+Desc. | OUC | 38.27 | **57.79** | 39.65 | 39.96 | **60.91** | 41.35 | 35.37 | **52.82** | 36.95 |
| | | | Tool | 44.00 | 66.49 | 46.12 | 42.77 | 66.06 | **44.82** | 42.12 | 65.44 | 42.45 |

Table 4: **Model performance on Ego4D SCOD. OD:** pure object detection. **Instr:** grounding with instructions. We highlight best OUC performance in **RED** for and best Tool performance in **GREEN**.

### 4.1.3 Results

Table 4 summarizes the overall model performance on Ego4D SCOD task. Even using the ground truth phrases, GLIP's zero-shot performance is significantly worse than pure OD baselines, implying that many of the SCOD objects are uncommon to its original training distribution. Generally, the instruction grounded performance (**Instr.**) are all better than the pure OD baselines, even with using the whole instruction sentence as the grounding phrase. The significant performance gaps between our models and the VidIntern baseline verifies that visual-only models can be much benefit from incorporation of textual information (should they be available) for the active object grounding task.

Particularly for OUC, with vanilla GPT extractions we can almost match the performance using the ground truth phrases, where the both the conditional and definition symbolic knowledge further improve the performance. Notice that condition knowledge by itself is more useful than the definition, and would perform better when combined. We also ablate a row excluding the per-object aggregation mechanism so that the conditional knowledge is simply utilized as a contextualized suffix for an instruction, which indeed performs worse, especially for the post-frames. As implied in Table 4, best performance on Tool is achieved using the ground truth phrases, leaving room for improvement on more accurate extractions and search of better suited symbolic knowledge.

| Method | Top-K | Post-Frame↑ | | OD Metric |
|---|---|---|---|---|
| | | AP | AP50 | AP75 |
| Track from GT PNR | 1 | 20.36 | 41.15 | 17.78 |
| Track from Pred. PNR | 1 | 10.21 | 21.27 | 8.63 |
| GPT+Conds.+Desc. | 1 | **29.85** | **43.53** | **31.45** |

Table 5: **PNR to Post OUC tracking ablation study.** Since tracking module only produce a single box for each frame, we report the top-1 performance of our grounding model. (Normally COCO API reports max 100 detection boxes.)

However, one may raise a natural question: if the OUC/Tool can be more robustly localized in the PNR frame, would a tracker improve the post-frame performance over our grounding framework? We thus conduct an ablation study using the tracker in Section 4.2 to track from PNR-frames using either the ground truth box and our model grounded box to the post-frames. Results in Table 5 contradicts this hypothesis, where we find that, due to viewpoint variations and appearance differences induced by the state change, our grounding model is significantly more robust than using tracking.

### 4.1.4 Qualitative Inspections

Figure 6 shows six different examples for in-depth qualitative inspections. It mainly shows that, generally, when the models grounding with the symbolic knowledge outperforms the ones without, the provided symbolic knowledge, especially the conditional knowledge, plays an important role

| FPV OD | BBox Rank. | Method | OPE-Det↑ | | OPE↑ | | Std. OD Metric↑ | | |
|---|---|---|---|---|---|---|---|---|---|
| | | | SS | NPS | SS | NPS | AP | AP50 | AP75 |
| HiC | MDNet | LTMU-H | 0.267 | 0.261 | 0.505 | 0.520 | — | — | — |
| HiC | MDNet | TbyD-H | 0.047 | 0.018 | 0.433 | 0.455 | — | — | — |
| Swin-L + DINO | VidIntern | — | 0.341 | 0.340 | 0.526 | 0.541 | 29.49 | 41.47 | 30.85 |
| GLIP-L | GLIP-L | Full-Instr. | 0.355 | 0.361 | 0.521 | 0.537 | 38.51 | 60.06 | 40.17 |
| | | SRL-ARG1 | 0.373 | 0.377 | 0.528 | 0.544 | 40.00 | 60.52 | 40.96 |
| | | GT-SRL-ARG1 | 0.383 | 0.390 | 0.531 | 0.548 | 42.35 | 61.41 | 44.27 |
| | | GPT | 0.379 | 0.389 | 0.529 | 0.545 | 41.85 | 61.89 | 43.46 |
| | | GPT+Desc. | 0.402 | 0.409 | 0.528 | 0.543 | 41.40 | 60.70 | 43.17 |
| | | GPT+Conds. | 0.412 | 0.422 | **0.541** | **0.557** | **45.90** | **67.26** | **47.94** |
| | | GPT+Desc.+Conds. | **0.413** | **0.424** | 0.539 | **0.557** | 43.49 | 64.34 | 45.43 |

Table 6: **Model performance on TREK-150.** OPE denotes One-Pass Evaluation (Dunnhofer et al., 2022) and OPE-Det is a variant to OPE where each tracker is initialized with its corresponding object detector prediction on the first frame. Success Score (SS) and Normalized Precision Score (NPS) are standard tracking metrics.

## 4.2 TREK-150

### 4.2.1 Experimental Setups

**Protocols.** TREK-150's official evaluation protocol is **One-Pass Evaluation (OPE)**, where the tracker is initialized with the ground-truth bounding box of the target in the first frame; and then ran on every subsequent frame until the end of the video. Tracking predictions and the ground-truth bounding boxes are compared based on IOU and distance of box centers. However, as the premise of having ground-truth bounding box initialization can be generally impractical, a variant of **OPE-Det** is additionally conducted, where the first-frame bounding box is directly predicted by our trained grounding model (grounded to the instructions).

**Evaluation Metrics.** Following Dunnhofer et al. (2022), we use common tracking metrics, *i.e.*, Success Score (SS) and Normalized Precision Score (NSP), as the primary evaluation metrics. In addition, we also report standard OD metric (APs) simply viewing each frame to be tracked as the localization task, as an alternative reference.

### 4.2.2 Baselines.

We adopt the best performing framework, the LTMU-H, in the original TREK-150 paper as the major baseline. LTMU-H integrates an fpv (first-person-view) object detector (HiC (Shan et al., 2020b)) into a generic object tracker (LTMU (Dai et al., 2020)), which is able to *re-focus* the tracker on the active objects after the (tracked) focus is lost (*i.e.*, identified by low tracker confidence scores).

Following the convention of utilizing object detection models to improve tracking (Feichtenhofer et al., 2017), we focus on improving object tracking performance by replacing the HiC-detector with our knowlede-enhanced GLIP models. We substi-

tute the HiC-detector for all 8 GLIP-based models and the VideoIntern baseline trained on the SCOD task and perform a zero-shot knowledge transfer (directly from Ego4D SCOD).[10]

### 4.2.3 Results

Table 6 summarizes the performance on TREK-150. Our best GLIP model trained using GPT-extracted objects and symbolic knowledge outperforms the best HiC baseline by over 54% relative gains in the SS metric and over 62% relative gains in the NPS scores for the OPE-D task. It also outperforms the VideoIntern baseline by 16-18% relative gains in SS/NPS and even the GLIP-(GT-SRL-ARG1) model by 7-9% relative gains on both metrics. This demonstrates the transferability of our OUC grounding model in fpv-tracking. For the OPE task with ground-truth initializations, the gains provided by our GLIP-GPT models over LTMU-H narrow to 7-8% relative gains across both metrics while still maintaining a lead over all other methods. This shows that the model is still able to better help the tracker re-focus on the OUCs although the overall tracking performance is more empirically bounded by the tracking module.

## 5 Related Works

**Egocentric Vision.** Egocentric vision has recently attracted research attentions thanks to advancements in smart wearable devices and robotics. Datasets used in this work, Ego4D (Grauman et al., 2022) and Epic-Kitchens (Damen et al., 2022, 2018; Dunnhofer et al., 2022) are two representative large-scale collections of egocentric videos

---

[10]Mainly because: (1) The general bounding box annotations in Epic-Kitchens videos are *machine annotated*, and (2) we believe model learned from Ego4D's more general visual domains should transfer well to kitchen activities.

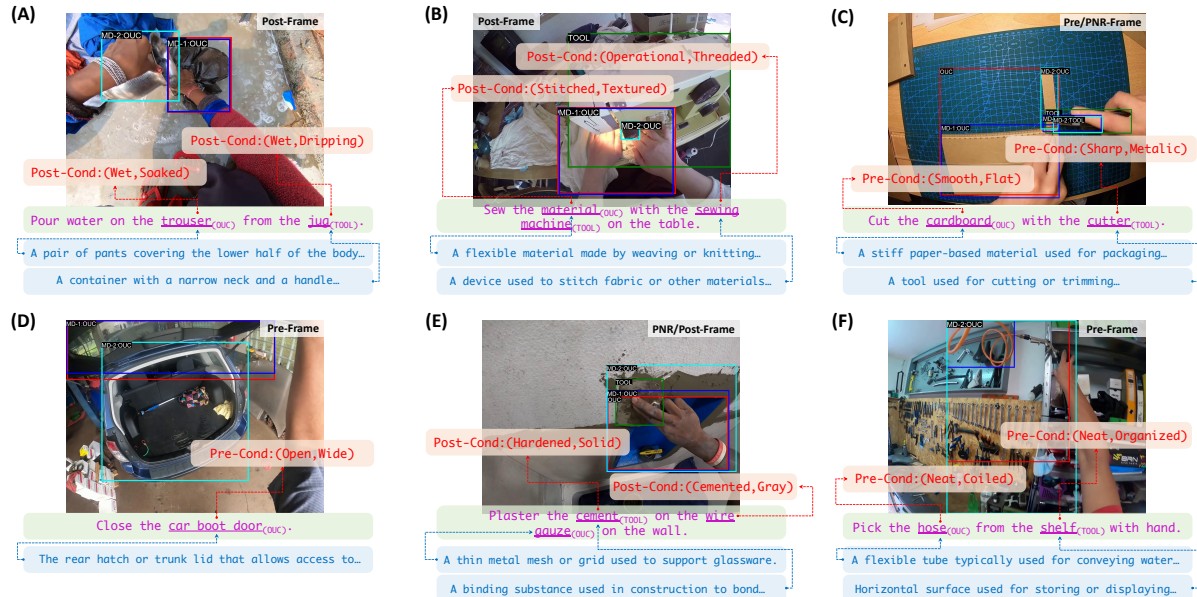

Figure 6: **Qualitative inspections**, mainly on the effectiveness of the GPT generated symbolic knowledge. Bounding box color code: Ground truth boxes, Models with uses of symbolic knowledge (MD-1), *i.e.*the **GPT+Conds.+Desc.**; Models without uses of symbolic knowledge (MD-2), *i.e.*, the vanilla **GPT**.

recording tasks performed by the camera wearers. Other existing works have also investigated egocentric vision in audio-visual learning (Kazakos et al., 2019), object detection with EgoNet (Bertasius et al., 2017; Furnari et al., 2017), object segmentation with eye-gazes (Kirillov et al., 2023) and videos (Darkhalil et al., 2022).

**Action-Object Knowledge.** The knowledge of objects are often at the center of understanding human actions. Prior works in both NLP and vision communities, have studied problems such as tracking visual object state changes (Alayrac et al., 2017; Isola et al., 2015; Yang et al., 2022), understanding object manipulations and affordances (Shan et al., 2020a; Fang et al., 2018), tracking textual entity state changes (Branavan et al., 2012; Bosselut et al., 2018; Mishra et al., 2018; Tandon et al., 2020), and understanding textual pre-/post-conditions from action instructions (Wu et al., 2023). While hand-object interactions (Shan et al., 2020a; Fu et al., 2022) are perhaps one of the most common object manipulation schemes, the objects undergoing change may not be directly in contact with the hands (see Figure 2). Here additional textual information can aid disambiguating the active object during localization and tracking. In this spirit, our work marries the merits from both modalities to tackle the active object grounding problem according to specific task instructions, and utilize action-object knowledge to further improve the models.

**Multimodal Grounding.** In this work, we adopt the GLIP model (Li* et al., 2022; Zhang* et al., 2022) for its compatibility with our settings and the joint inference framework, which indeed demonstrate significant improvements for the active object grounding task. There are many related works for multimodal grounding and/or leveraging language (LLMs) to help with vision tasks, including (but not limited to) Grounding-DINO (Liu et al., 2023), DQ-DETR (Liu et al., 2022), ELEVATER (Li* et al., 2022), phrase segmentation (Zou* et al., 2022), visually-enhanced grounding (Yang et al., 2023), video-to-text grounding (Zhou et al., 2023), LLM-enhanced zero-shot novel object classification (Naeem et al., 2023), and multimodal object description generations (Li et al., 2022, 2023).

## 6   Conclusions

In this work, we approach the active object grounding task leveraging two narrated egocentric video datasets, Ego4D and Epic-Kitchens. We propose a carefully designed prompting scheme to obtain useful action-object knowledge from LLMs (GPT), with specific focuses on object pre-/post-conditions during an action and its attributional descriptions. Enriching the GLIP model with the aforementioned knowledge as well as the proposed per-object knowledge aggregation technique, our models outperforms various strong baselines in both active object localization and tracking tasks.

## 7 Limitations

We hereby discuss the potential limitations of our work:

**(1)** While we make our best endeavours to engineer comprehensive and appropriate prompts for obtaining essential symbolic action-object knowledge from large language models (LLMs) such as GPT, there are still few cases where the extracted objects are not ideal (see Table 1). Hence, our model performance could potentially be bounded by such limitation inherited from the LLM ability to fully and accurately comprehend the provided instructions. Future works can explore whether more sophisticated in-context learning (by providing examples that could be tricky to the LLM) would be able to alleviate this issue. Alternatively, we may utilize LLM-self-constructed datasets to fine-tune another strong language models (such as Alpaca (Taori et al., 2023)) for the object extraction task. On the other hand, incorporating high-level descriptions of the visual contexts using off-the-shelf captioning models could also be explored to make the LLM more *situated* to further improve the efficacy of the extracted knowledge.

**(2)** As this work focuses on action frames in first-person egocentric videos (from both Ego4D (Grauman et al., 2022) and Epic-Kitchens dataset (Damen et al., 2018; Dunnhofer et al., 2022) datasets), the underlying learned model would obviously perform better on visual observations/scenes from first-person viewpoints. While we hypothesize that, unless heavy occlusion and drastic domain shifts (of the performed tasks and/or objects involved) occur, the learned models in this work should be able to transfer to third-person viewpoints, we have not fully tested and verified such hypothesis. However, if applied properly, the overall framework as well as the utilization of LLMs for action-object knowledge should be well-generalizable regardless of the viewpoints.

**(3)** There is more object- and action-relevant knowledge that could be obtained from LLMs, such as spatial relations among the objects, size difference between the objects, and other subtle geometrical transitions of the objects. During experiments, we attempted to incorporate spatial and size information to our models. However, experimental results on the given datasets did not show significant improvement. Thus we omitted them from this work. We hope to inspire future relevant research along this line to further exploit other potentially useful knowledge.

## 8 Ethics and Broader Impacts

We hereby acknowledge that all of the co-authors of this work are aware of the provided *ACL Code of Ethics* and honor the code of conduct. This work is mainly about understanding and localizing the key objects undergoing major state changes during human performing an instructed or guided action, which is a crucial step for application such as on-device AI assistant for AR or VR devices/glasses.

**Datasets.** We mainly use the state-change-object-detection (SCOD) subtask provided by Ego4D (Grauman et al., 2022) grand challenge for training and (testing on their own tasks), and transfer the learned knowledge to TREK-150 (Dunnhofer et al., 2022) from Epic-Kitchens dataset (Damen et al., 2018). These datasets is not created to have intended biases towards any population where the videos are collected spanning across multiple regions around the world. The main knowledge learned from the dataset is mainly physical knowledge, which should be generally applicable to any social groups when conducting daily tasks.

**Techniques.** We propose to leverage both the symbolic knowledge from large-language models (LLMs) such as GPT to guide the multimodal grounding model and the visual appearances and relations among objects for localizing the object undergo changes. The technique should be generally transferable to similar tasks even outside of the domain concerned in this work, and unless misused intentionally to harmful data (or trained on), should not contain harmful information for its predictions and usages.

## Acknowledgments

Many thanks to I-Hung Hsu for his valuable feedback during our paper preparation, and to Liunian Harold Li for his tips on using his great work, GLIP models. This material is based on research supported by the Machine Common Sense (MCS) program under Cooperative Agreement N66001-19-2-4032 and the ECOLE program under Cooperative Agreement HR00112390060, both with the US Defense Advanced Research Projects Agency (DARPA). The views and conclusions contained herein are those of the authors and should not be interpreted as necessarily representing DARPA, or the U.S. Government.

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

# A GPT Prompt Engineering Details

## A.1 GPT Prompts

Below are the prompts used in the GPT entity and symbolic knowledge extraction pipeline as described in Figure 5.

```
# GPT Prompt Set

# Identify OUC & Tool:
GPT Prompt 1 = "From the action [{}],
    please extract the follow objects
    exactly as they are written: (a) The
     single object being manipulated (b)
     The one single tool used to
    manipulate (a). Please do not
    explain anything in you answer.
    Please only output one object for
    each part. If you cannot find either
     (a) or (b), please simply output [
    None] for that field and print
    nothing else."

# Precise Semantic Grounding (for both
    OUC and Tool):
GPT Prompt 2 = "Please output an exact
    subpart of the sentence [{}] where
    the object [{}] is referred to.
    Please remove leading verbs in your
    answer. Please output only the exact
     sentence subpart and nothing else.
    Please make sure that you answer can
     be exactly found in the sentence
    [{}]. If question is not valid,
    please simply output [None] and
    nothing else. Please do not output
    any explaination of your answer. "

# State Change Forcast
GPT Prompt 3 = "From the first person
    view of the action [{}], would the
    object [{}] undergo significant
    change in its visual appearance? If
    so, please output [yes] in the first
     line and If not, please output [no]
     in the first line. If the answer is
     yes, then on the second line,
    please simply print one or two
    visually recognizable adjectives to
    describe the appearance of the
    object before the state change, on
    the third line, please simply print
    a few (<7) words that visually
    describe the object after the state
    change"

# Object Description
GPT Prompt 4 = "In one sentence, please
    define the object [{}] visually"
```

## A.2 GPT Pipeline

The GPT prompts are used in the order above. First, we plug in the instructional caption to GPT Prompt 1 and generate a tentative OUC-Tool pair. If either entity is not found, GPT would return None and

future queries will ignore that entity.

Due to GPT's tendency to paraphrase and provide unwanted chain-of-thought explanations, we further pass GPT outputs for OUC and/or tool through an additional query aimed to produce exact grounding if it is not contained in the caption. Emperically this query has a very high chance of producing a caption-groundable answer if exist. In the unlikely event that no groundable OUC is found after running GPT Prompt 2, we resort to using SRL [ARG1] as the OUC. On the other hand, if no groundable Tool is found after running GPT Prompt 2, we set the Tool to None.

At this point, we run GPT Prompts 3 and 4 on the OUC and Tool (if exist). GPT results are parsed and stored if following the designated formats.

## A.3 GPT Situated Role

For a small randomly selected subset of Ego4d prompts (220 samples) in the human evaluation of GPT generated answers for pre/post conditions (Table 2), we evaluated results both with and without the situated role specification "From the first-person view." in Table 7.

| Prompt Style | Pre Cond. | | Post Cond. | |
|---|---|---|---|---|
| | Textual | Visual | Textual | Visual |
| wo/ Situated Role | 70.9 | 75.0 | 61.4 | 64.5 |
| w/ Situated Role | 86.4 | 82.3 | 75.5 | 71.8 |

Table 7: **GPT prompt comparison on Ego4D subset.** Here "Textual" refers to "textual correctness": i.e. "based on text and common sense alone, does the GPT conds./desc. make sense?"; whereas "Visual" refers to "visual correctness": i.e. "does the GPT conds./desc. match what is shown in the image?"

We observed that when provided with additional situated viewpoint specification, GPT results displayed a significant increase in alignment with human judgment, thus it was incorporated into our final prompt.

# B Details of Modeling & Learning

## B.1 Narration Processing

**Ego4D.** Ego4D's original annotated narrations are of third-person descriptive tone, *e.g.*, *"#C cuts the vegetables with the knife."*, where the symbol of #C indicates the camera wearer. Since this pattern is universally applicable to all the available narrations of the videos, we perform a deterministic string transformation stripping these special indi-

cators of camera wearers or other human characters, followed by a simple conversion to change the third-person singular verb to first-person verb. The rest of the pronouns are also deterministically converted, so that the narration becomes a first-person imperative tone. Although we did not empirically find such a transformation make any significant impacts to the model performance, we conduct this transformation for GPT to more easily grasp the situated roles (given to it) as well as conform to the main motivation of this work better.

**Epic-Kitchens.** For TREK-150 where its videos basically belong to Epic-Kitchens, the original released narrations are already in an imperative instructional tone. In light of this, we simply adopt them as the task instructions seamlessly without needing to perform any modifications.

## B.2 Training & Implementation Details

**Training Details.** For GLIP series of models, we simply truncate the textual inputs (the action instruction texts) at maximum 256 tokens, ensuring all of the textual inputs are below such maximum length even with additional symbolic knowledge is incorporated. The hyperparameters are manually tuned against an Ego4D SCOD validation set, and the checkpoints used for testing are selected by the best performing ones on such set.

All the models in this work are trained on 2-4 Nvidia A100/A6000 GPUs[11][12] on a Ubuntu 20.04.2 operating system. The hyperparameters for each model are manually tuned against the development datasets, and the checkpoints used for testing are selected by the best performing ones on such held-out development sets.

**Implementation Details.** The implementations of the transformer-based models are extended from the HuggingFace[13] code base (Wolf et al., 2020), and our entire code-base is implemented in PyTorch.[14]

## B.3 Hyperparameters

We train our models until performance convergence is observed on the held-out development set (split from the original train set from Ego4D SCOD subtask). The training time is roughly 12-14 hours, spanning 10-15 training epochs. We list all the hyperparameters used in Table 8. The basic hyperparameters such as learning rate, batch size, and gradient accumulation steps, are kept consistent for models based off the same architecture (for baselines and our GLIP-L model series). All of our models adopt the same search bounds and ranges of trials as in Table 9.

---

[11]https://www.nvidia.com/en-us/data-center/a100/
[12]https://www.nvidia.com/en-us/design-visualization/rtx-a6000/
[13]https://github.com/huggingface/transformers
[14]https://pytorch.org/

| Models | Batch Size | Initial LR | # Training Epochs | Gradient Accumulation Steps | # Params |
|---|---|---|---|---|---|
| VideoIntern Baselines | 8 | $1 \times 10^{-3}$ | 10 | 1 | 218M |
| GLIP-L Series | 4 | $1 \times 10^{-3}$ | 10 | 1 | 429M |

Table 8: **Hyperparameters in this work:** *Initial LR* denotes the initial learning rate. All the models are trained with Adam optimizers (Kingma and Ba, 2015). We include number of learnable parameters of each model in the column of *# params*.

| Type | Batch Size | Initial LR | # Training Epochs | Gradient Accumulation Steps |
|---|---|---|---|---|
| **Bound (lower–upper)** | 2–8 | $1 \times 10^{-3}$–$1 \times 10^{-5}$ | 5–15 | 1 |
| **Number of Trials** | 2–4 | 2–3 | 2–4 | 1 |

Table 9: **Search bounds** for the hyperparameters of all the models.