# OpenReview forum: "Localizing Active Objects from Egocentric Vision with Symbolic World Knowledge"
_EMNLP/2023/Conference — EMNLP 2023 Main_

### Official Review · Reviewer_h4o5 · 2023-07-26

**Soundness:** 4

**Excitement:**

3: Ambivalent: It has merits (e.g., it reports state-of-the-art results, the idea is nice), but there are key weaknesses (e.g., it describes incremental work), and it can significantly benefit from another round of revision. However, I won't object to accepting it if my co-reviewers champion it.

**Missing References:**

Muhammad Ferjad Naeem, Muhammad Gul Zain Ali Khan, Yongqin Xian, Muhammad Zeshan Afzal, Didier Stricker, Luc Van Gool, Federico Tombari; I2MVFormer: Large Language Model Generated Multi-View Document Supervision for Zero-Shot Image Classification. Proceedings of the IEEE/CVF Conference on Computer Vision and Pattern Recognition (CVPR), 2023, pp. 15169-15179

**Paper Topic And Main Contributions:**

This paper considers the problem of actively grounding task instructions from an egocentric view. More concretely, authors deal with the problem of detecting and tracking objects undergoing change and tools (other objects which might be used to change the state of the main object). Previous works have mainly focused on the vision modality. This work tries to leverage also the textual modality, extracting symbolic knowledge from LLMs and training an extension of an open-vocabulary object detector (GLIP) to improve its grounding capabilities. The proposal is evaluated on two egocentric video datasets, namely Ego4D and Epic-Kitchens.

**Questions For The Authors:**

A- In Section 2, you present the task itself. You mention that you rephrase the narrations as imperative instructions. I don't really see the need for that, but I may be missing something. How would the evaluation of your method, or even the method itself, change if you used the original narrations?
B- You use ChatGPT, a closed model that we don't even know where was trained, to generate symbolic knowledge. Did you check whether ChatGPT is contaminated by Ego4D and Epic-Kitchens narrations? Have you considered the usage of open source models such as Llama and similars?
C- In Section 3.2 you mention that "Interestingly, we empirically find it beneficial to situate GPT with a role, e.g., "From the first-person view."". Can you provide more details on the empirical process? Was it qualitative or quantitative?

**Reasons To Accept:**

A- The idea of using LLMs to extract symbolic knowledge from textual instructions to improve object detection and tracking is very interesting.
B- The proposed method (extended GLIP) can also be used for further multimodal or vision-based tasks.
C- The obtained results show that leveraging textual information (in various ways) improves over purely visual methods.

**Reasons To Reject:**

A- The presentation of the paper could be improved significantly, specially concerning the figures and their captions (they are not very descriptive) and the tables/captions. For example: the caption of Figure 1 does not describe the figure itself; it is a shorter repetition of the abstract. It would be more helpful to describe what the figure shows. The caption of Figure 4 is also very brief, when the depicted diagram is not easy to understand. Table 1: each row is different but it is not explained in the caption. For instance, what is "Defs" in the first row? Table 2 shows actually two different tables, and nothing is reported in the caption about the meaning of "Textual" and "Visual". Finally, Table 3, which is the main table of the paper, is so dense that it is very difficult to find trends and extract conclusions. I think this table should be split.
B- Although the idea of using LLMs to extract symbolic knowledge and leveraging that knowledge to improve visual tasks is very interesting, it is not novel, from my point of view. Check the following paper, for example:
Muhammad Ferjad Naeem, Muhammad Gul Zain Ali Khan, Yongqin Xian, Muhammad Zeshan Afzal, Didier Stricker, Luc Van Gool, Federico Tombari; I2MVFormer: Large Language Model Generated Multi-View Document Supervision for Zero-Shot Image Classification. Proceedings of the IEEE/CVF Conference on Computer Vision and Pattern Recognition (CVPR), 2023, pp. 15169-15179
Even though the application is different and the methods differ, I think the main idea is quite similar. I still believe that the instantiation of the idea in this work is interesting, but I have to admit that the idea itself is not novel.
C- Some doubts about the obtained results: specially for Table 3, authors do not specify how many executions they performed to obtain those results. At least three executions with the average and standard deviation should be reported to assess whether the obtained differences among models is significant. If those three (or more) executions cannot be performed, it should be justified. Similarly, for Table 2, there are missing details. Those are human evaluation results, but authors do not mention how many evaluators were involved, their background, their relation with the research team, etc. I know those are "additional results" and not the main target of the work, but I think we have to be more rigorous when presenting results (an Appendix could be added, for example, with the missing details).

**Reproducibility:**

4: Could mostly reproduce the results, but there may be some variation because of sample variance or minor variations in their interpretation of the protocol or method.

**Reviewer Confidence:**

4: Quite sure. I tried to check the important points carefully. It's unlikely, though conceivable, that I missed something that should affect my ratings.

**Typos Grammar Style And Presentation Improvements:**

The paper contains many typos. Here is a list (not exhaustive; please, review the paper for typos):

line 163: challengeonly -> challenge only
Figure 5: State Change Forcast -> State Change Forecast
line 312: can be deterministic constructed -> deterministically constructed

---

> ### Author Rebuttal · Authors · 2023-08-29
>
> We thank Reviewer h4o5 for the helpful comments and detailed questions! We are encouraged that the reviewer finds **our main idea very interesting**, and that the **extensive experimental studies justify its effectiveness**. We are particularly delighted that the reviewer acknowledges our framework being **generalizable** and **can be applied to further multimodal research**.
>
> Please find your suggestions and concerns addressed below:
>
> ---
>
> **[W1] Paper presentation improvements**
>
> Sorry for the confusions and we really appreciate all the paper presentation suggestions.
> We hereby address the concerns and elaborate on how we would revise our paper. We believe these revisions can be easily performed in the final version and hope the reviewer can re-evaluate this point as a reason to reject our work:
>
> - **Figure 1.** Revised caption: *”**Active object grounding** is the task of localizing the active objects undergoing state change (OUC). In this example action instruction "cut the pawpaw into half with the knife", the AI assistant is required to firstly infer the OUC (**pawpaw**) and the Tool (**knife**) from the instruction, and then localize them in the egocentric visual scenes throughout the action trajectories. Symbolic knowledge including pre/post conditions and object descriptions can bring additional information to facilitate the grounding.”*
>
> - **Figure 4.** We will condense L216-220 to enrich the description of the contrastive learning (left part), L276-279 to the additional object-knowledge grounding (right part), and L285-288 to the frame type prediction part.
>
> - **Table 1.** Defs. (which stands for Definitions) and Desc. (Descriptions of an object provided by the LLM) are equal. It was an editorial error which we will fix. Thanks for pointing it out! This table is a qualitative inspection of representative cases where the LLM fails to capture the desired knowledge from the action instructions. For clarity, we will change the Table 1 caption to the following: *”Table 1: Qualitative Analysis of GPT Knowledge Extraction: Examples of cases where GPT extracted symbolic knowledge are wrong or conflict with Ego4D annotations. Here the GPT-extracted or dataset-annotated knowledge are displayed in GREEN if they match human analysis and RED otherwise.”*
>
> - **Table 2.** As explained in Section 3.2 (L258-260), *“Textual”* refers to **”textual correctness”**: *i.e.* *"based on text and common sense alone, does the GPT conds./desc. make sense?"*; whereas *“Visual”* refers to **”visual correctness”**: *i.e.* *“does the GPT conds./desc. match what is shown in the image?"* For more clarity, we will move this explanation to the image caption in the final version.
>
> - **Table 3.** In our final version we will split the table into two, where one contains the major results comparing our best model combinations with the baselines and other utilization of the action instructions; while the other mainly left for ablating the types of knowledge to be jointly inferred on. We also consider reporting only primary AP and AP50 results for the main paper and moving AP75 to the appendices if that brings better readability.
>
> Overall, we thank and appreciate all the paper presentation feedback and we will surely incorporate them into our final version!
>
> ---
>
> **[W2] Novelty of our work**
>
> Thanks for the related work! Ever since LLM showed remarkable performance on many general tasks, the incorporation of LLM into a multimodal system has drawn large attention from the community. While we acknowledge that the *“incorporation”* (itself) of an LLM is not novel, the actual usage of LLMs and types of knowledge carried out for different applications distinguishes each work.
>
> Particularly for the I2MVFormer work, LLMs are used to derive multiple *“views”* (*i.e.*, **static descriptions**) of an object class following a Wiki-style.
> While this aligns at a high-level with the generic-description (desc.) sub-component of our framework, our main novelty also lies in the utilization of **action- and state- dependent pre- and post-conditions**. These conditions are especially beneficial for **multi-frame action videos** and other **real-time vision applications** where the **state changes of objects** (visual contexts, appearances, etc.) are **spatiotemporally action-dependent beyond their generic attributes [1,2]**.
>
> Our experimental results (Table 3 and 5) also support this claim as *GPT+Conds.* significantly outperforms *GPT+Desc.* in nearly every evaluation category (and achieves best results in Ego4D SCOD when combined).
>
> Furthermore, the inference we propose is applicable to not just one object at a time but to **multiple objects involved in the human tasks** (*i.e.*, in this work, both Tools and OUCs). And hence, to make the derived symbolic knowledge concise but informative, we made efforts to devise an effective prompting scheme that is robust (tested on different versions of OpenAI GPT-series) to be further processed and parsed to help the grounding modules.
>
> We will include the suggested work into the multimodality related work section of ours. Thanks for the suggestion!
>
> [1] Souček, Tomáš, et al. "Look for the change: Learning object states and state-modifying actions from untrimmed web videos." *CVPR* 2022
>
> [2] Isola, Phillip, Joseph J. Lim, and Edward H. Adelson. "Discovering states and transformations in image collections." *CVPR* 2015
>
> ---
>
> **[W3] Supporting details for experimental results**
>
> We mainly reported the best results obtained from each model. While during the submission period we trained key models on three or more different random seeds, due to time and computational resource constraints, we did not have the luxury to sweep the entire table for the multi-trials.
> Here we supplement the performance mean and std from the major model comparisons:
>
> | **Model** |           | **Pre** | **Pre** | **Pre** | **PNR** | **PNR** | **PNR** | **Post** | **Post** | **Post** |
> | :-----:| :-----: | :-----: | :-----: | :-----: | :-----: | :-----: | :-----: | :-----: | :-----: | :-----: |
> |           |           | **AP** | **AP50** | **AP75**  | **AP** | **AP50** | **AP75** | **AP** | **AP50** | **AP75** |
> | Full-Instr. | Mean | 33.07 | 52.33 | 34.04 | 34.53 | 55.46 | 35.33 | 30.90 | 48.29 | 32.12 |
> | Full-Instr.  | Std. | 0.44 | 0.50 | 0.52 | 0.55 | 0.73 | 0.60 | 0.29 | 0.66 | 0.29 |
> | GPT | Mean | 37.24 | 55.61 | 38.80 | 39.16 | 59.18 | 40.18 | 34.75 | 51.18 | 36.53 |
> | GPT  | Std. | 0.18 | 0.36 | 0.14 | 0.13 | 0.23 | 0.12 | 0.02 | 0.12 | 0.13 |
> | GT-SRL-ARG1 | Mean | 37.45 | 56.28 | 38.92 | 39.10 | 59.38 | 40.19 | 34.65 | 51.32 | 36.25 |
> | GT-SRL-ARG1  | Std. | 0.35 | 0.17 | 0.54 | 0.43 | 0.16 | 0.42 | 0.33 | 0.03 | 0.40 |
> | GPT+Conds. | Mean | 38.44 | 57.33 | 39.91 | 39.99 | 60.38 | 41.35 | 35.28 | 52.06 | 37.01 |
> | GPT+Conds. | Std. | 0.21 | 0.23 | 0.17 | 0.20 | 0.09 | 0.21 | 0.15 | 0.14 | 0.22 |
> | GPT+Conds.+Desc. | Mean | 38.17 | 57.32 | 39.69 | 39.92 | 60.60 | 41.24 | 35.44 | 52.67 | 37.02 |
> | GPT+Conds.+Desc. | Std. | 0.19 | 0.30 | 0.29 | 0.14 | 0.31 | 0.24 | 0.28 | 0.34 | 0.44 |
>
> The results show close alignment with performance trends reported in the main paper.
>
> We asked two trained (on the understanding of the pre- and post-conditions of an action-object interaction) non-co-author internal lab members to perform the human evaluations in Table 2, by sending them spreadsheets containing the original action instructions, corresponding LLM extractions, and the LLM-generated symbolic knowledge, where the human evaluated (*binary good/bad*) scores are averaged across their evaluations.
>
> We appreciate the suggestions on the additional details of our studies, we will incorporate them into either the main paper or a new section in the appendix.
>
> ---
>
> **[Q1] Imperative formats**
>
> We rephrase task narrations to imperative form simply to follow the motivation of situating the assistant AI into the real-world application scenario. You are correct that our method does not (and should not) show any significant difference in the two formats of narrations.
>
> ---
>
> **[Q2] ChatGPT data contamination (symbolic knowledge extraction sub-task)**
>
> In our framework, we task ChatGPT with extracting the following knowledge from task instructions: the objects undergoing state change (OUCs), the tools that facilitate such actions, the pre- and post-conditions of each object, and the descriptions of each object. It is important to note that the Ego4D dataset does not contain annotations for any symbolic knowledge, and they did not previously exist anywhere until we generated them. Therefore, it is impossible for ChatGPT to have seen exact answers to our symbolic knowledge extraction questions even if it had access to all Ego4D annotations. What's more, the Ego4D annotations themselves are by no means a well-structured resource available online, and therefore extremely unlikely to be acquired and effectively trained on. Considering the above points, we do not believe that ChatGPT can effectively exploit the Ego4D narrations and associated symbolic knowledge in training time.
>
> Our method is not limited to the type of the LLMs, and hence high performing Llama family models can be suitable candidates for replacing ChatGPT in our knowledge extraction component. In any case, our framework is concerned only with the LLM's ability to follow our instructions and perform knowledge extraction.
>
> ---
>
> **[Q3] Prompt specification of first-person view**
>
> **Some background:**
> - Here the task is for GPT to determine whether the object being questioned goes through visual state change from the action performer’s egocentric view and if so, provide visually identifiable pre/post conditions, thus we use this prompt phrase to provide GPT with viewpoint information and exploit its (*simulated*) visual commonsense knowledge (full prompt in Appendix A.1 L954-970).
> - This similar phenomenon is also observed in some concurrent works/blog posts [3,4,5] when utilizing LLMs via a role-playing fashion.
>
> **The empirical process was both qualitative and quantitative**:
> - **Qualitatively**: given the instruction "Lay the trousers on the ironing board" and asked whether “the ironing board” goes through any visual state change, GPT (specified first-person view) would respond with “Yes” (as the ironing board becomes covered by the trousers in the action performer’s egocentric view) and go on to provide the post condition “blocked, covered”. For the same example without specifying "From the first-person view.", GPT would respond with “No” and provide no post conditions.
> - **Quantitatively**: For a small subset of Ego4d prompts (220 samples) in the human evaluation of GPT generated answers for pre/post conditions (Table 2), we evaluated results both w/wo the specification "From the first-person view.":
> | **Prompt Style** |  Pre Cond.| |  Post Cond.| |
> | :-----:   | :-----: | :-----: | :-----: | :-----: |
> | | **Textual** | **Visual** | **Textual** | **Visual** |
> | wo/ First-Person | 70.9 | 75.0 | 61.4 | 64.5 |
> | w/ First-Person | 86.4 | 82.3 | 75.5 | 71.8 |
>   - We observed that when provided with additional viewpoint specification, GPT results displayed a significant increase in alignment with human judgment, thus it was incorporated into our final prompt.
>
> [3] Ding, Yan, et al. "Leveraging Commonsense Knowledge from Large Language Models for Task and Motion Planning." *RSS 2023 Workshop* 2023
>
> [4] Shanahan, Murray, Kyle McDonell, and Laria Reynolds. "Role-Play with Large Language Models." *arXiv preprint* 2023
>
> [5] Babar M Bhatti. “The Art and Science of Crafting Effective Prompts for LLMs”, *Medium Blog Post* 2023
>
> ---
>
> **[Misc.] Missing references and typos**
>
> We will cite the suggested references appropriately in our final version. Specifically for the I2MVFormer paper, we will include the discussion mentioned above. Thanks again for the presentation suggestion, we will fix the typos and enrich the related works section accordingly.

---

### Official Review · Reviewer_5jwp · 2023-08-05

**Soundness:** 3

**Excitement:**

3: Ambivalent: It has merits (e.g., it reports state-of-the-art results, the idea is nice), but there are key weaknesses (e.g., it describes incremental work), and it can significantly benefit from another round of revision. However, I won't object to accepting it if my co-reviewers champion it.

**Missing References:**

1. Active Object Detection With Multistep Action Prediction Using Deep Q-Network, IEEE Transactions on Industrial Informatics 2019
2.Active object localization with deep reinforcement learning
3.Sequential voting with relational box fields for active object detection

**Paper Topic And Main Contributions:**

**Topic:** This paper aims to address the active object grounding, that is, to find the objects undergoing state-change in egocentric videos.

**Contribution:**
1. propose a prompting pipeline that integrates LLM to obtain the knowledge about the objects undergoing state-change
2. propose a joint inference framework that aggregates the pre- and post-condition information to find the objects undergoing state-change
3. evaluate the proposed framework on two egocentric video datasets

**Reasons To Accept:**

1.propose a  prompting pipeline that integrates the knowledge about the objects undergoing state-change from LLM
2.evaluate the proposed framework on two egocentric video datasets

**Reasons To Reject:**

1. The paper lacks comparison with existing active object detection methods, like[1,2,3] in  missing references or other object grounding methods
2. The comparisons with VidIntern is not fair. For the objects undergoing state-change detection, VidIntern does not take any caption or instruction as inputs, meanwhile, it detect the active object on the PNR not including Pre and Post frames.
3.The details of the epic-kitchen(trek-150) is not sufficient, such as the definition of pre and pose frames.

**Reproducibility:**

4: Could mostly reproduce the results, but there may be some variation because of sample variance or minor variations in their interpretation of the protocol or method.

**Reviewer Confidence:**

5: Positive that my evaluation is correct. I read the paper very carefully and I am very familiar with related work.

---

> ### Author Rebuttal · Authors · 2023-08-29
>
> We thank Reviewer 5jwp for the constructive feedback! Please find your suggestions and concerns addressed in detail below:
>
> ---
>
> **[W1] Comparisons with existing works [1,2,3]**
>
> We carefully reviewed the suggested related works and will hereby discuss their relations to our paper.
> We would like to firstly clarify a potential misunderstanding of the term *”active object”* and our newly-defined task *”active object grounding”*.
> In our paper (**L124-131 and L160-165**), we clearly define the task **active object grounding** as automatically localizing and tracking **object(s) undergoing state change (OUC)** (objects being passively manipulated by the agent) and **Tool(s)** used to enable **an instructed manipulation**. OUCs and Tools are in turn defined as *“active objects”*, which are not directly named and thus need to be extracted from the task instructions.
>
> - **Paper [1]**, while titled *“active object detection”*, is actually discussing the problem of trying to detect a **named object** via an ***"active"* mobile agent traversing a 3d environment** and taking pictures of the object from *different angles*. Here the word *“active”* is referring to the mobile nature of the agent and not the object. Unlike our work, here the agent is a mere observer and not in the view. The objects are not going through any form of state change.
>
> - **Paper [2]** also has *“active object detection”* in its title but is actually a work on trying to improve generic **named object detection** (on the Pascal VOC dataset) by **iteratively cropping the image** in an ***“active”* manner**. **Again here *“active”* does not refer to the object itself** but rather describes the authors’ method to improve a completely different task. Here the images are not egocentric and there is no agent involved. Similar to Paper [1], the objects here are not going through any form of state change.
>
> - **Paper [3]** defines *“active objects”* as objects **directly in contact** with the human hand. Recall from above that our definition of *“active object”* is *the object undergoing state change due to an instructed action*, and hence this object **is often not directly in contact with human hands**. The goal of Paper [3]’s task is to exploit the human hands’ role in providing location hints for object detection and dealing with partial object occlusion caused by hands blocking the camera's direct view of the object, which is completely different from our task’s goal. Nevertheless, we conduct an additional experiment by applying this work to our Ego4D SCOD active object grounding task (on OUC localization only). Within the short author response period, we have **reached out to the original authors** and **made our best efforts to re-implement** Paper [3]’s model. We worked closely with them to **fairly re-train and evaluate their model** on our dataset. In this process, **we provided Paper [3]’s framework with the annotated *ground truth* hand bounding boxes** (which already gives it an additional advantage over our method). We report its corresponding performances below:
>
> - **Overall Performance (%) on Ego4D SCOD**
> | **Pre** | **Pre** | **Pre** | **PNR** | **PNR** | **PNR** | **Post** | **Post** | **Post** |
> | :-----: | :-----: | :-----: | :-----: | :-----: | :-----: | :-----: | :-----: | :-----: |
> | **AP** | **AP50** | **AP75**  | **AP** | **AP50** | **AP75** | **AP** | **AP50** | **AP75** |
> | 0.008 | 0.021 | 0.000 | 0.198 | 0.330 | 0.330 | 0.005 | 0.026 | 0.001 |
>
> To reflect the performance shown above, we additionally analyze the IOUs between the predicted bounding boxes and the ground truth ones, for their mean/std and histogram statistics, which are reported below:
>
> - **Mean and Std of IOUs**
> | **Pre** | **Pre** | **PNR** | **PNR** | **Post** | **Post** |
> | :-----: | :-----: | :-----: | :-----: | :-----: | :-----: |
> | **Mean** | **Std** | **Mean**  | **Std** | **Mean** | **Std** |
> | 0.0484 | 0.0976 | 0.0488 | 0.0993 | 0.0478 | 0.0967 |
>
> - **IOU Histogram (%)**
> | **0.0-0.1**| **0.1-0.2** | **0.2-0.3** | **0.3-0.4** | **0.4-0.5** | **0.5-0.6** | **0.6-0.7** | **0.7-0.8** | **0.8-0.9** | **0.9-1.0** |
> | :-----: | :-----: | :-----: | :-----: | :-----: | :-----: | :-----: | :-----: | :-----: | :-----: |
> | 83.16 | 8.64 | 4.31 | 2.09 | 1.21 | 0.36 | 0.21 | 0.07 | 0.02 | 0.00 |
>
> Above results show that the performance of Paper [3]’s framework is **far below our lowest performing visual-only baseline** (*i.e.*, VidIntern) on the *active object grounding* task (as defined in our work) especially for Pre and Post frames. The low performance can mostly be attributed to the model’s **fundamentally different objective** to detect *the hand-held object* instead of the actual *object undergoing state change*. This result further highlights the difficulty and unique nature of our proposed task.
>
> ---
>
> **[W2] Comparisons with the VidIntern baseline**
>
> While we agree that VidIntern does not utilize any caption information, this is **precisely one of the motivations of our work (see L61-65 in the Introduction)** – **whether incorporating textual information can significantly benefit the *active object grounding* task**. This is a very practical setting for many real-world applications (*e.g.*, scenarios depicted in Introduction L41-56) where the textual task instructions are readily available (or uttered).
>
> It’s important to note that both datasets we experimented on provide such textual information to be used at inference time, *i.e.*, we did not collect or provide any additional annotations for our model. Thus, we believe that our comparison with VidIntern (as our strongest vision-only baseline) is indeed fair as **both frameworks are restricted to training only on dataset-designated resources**.
>
> Our experiments successfully demonstrated that enhancing vision-only object detection with multimodal joint inference can yield significant performance improvements on the Ego4D SCOD task.
> This result reinforces the emerging research direction of augmenting vision-only tasks with textual information, where effectively utilizing multiple available modalities to learn a stronger model compared to its unimodal counterparts are also well-studied in [4, 5, 6, 7].
>
> Regarding concerns related to evaluating Pre/Post frames in addition to PNR frames, we would like to note that the Ego4D SCOD Track provides official train/val data for all frames including Pre/PNR/Post.
> The recorded performance of VidIntern on Pre/Post frames is its best performance after being trained on all frames just like our models, thus making certain that our comparison is fair.
>
> While our model outperforms VidIntern on PNR frames by a significant amount (3.87 AP50), the performance gap further increases for Pre and Post frames. This can be attributed to the fact that in PNR frames the OUCs are closer to the human actors (and their hands), making it easier for pure vision models to leverage this spurious correlation and disambiguate the OUC without actually understanding which object is undergoing state change.
>
> Lastly, GLIP [8, 9] itself is one of the state-of-the-art phrase/object  grounding (*requires knowing which phrase(s) to ground **beforehand***) and open-vocabulary object detection models in public.
> All our model variants without using the symbolic knowledge in Table 3, are **exactly baseline comparisons with the generic GLIP**, only that we ablate the **automated** *phrase-to-ground generation* with our proposed GPT extractions against heuristic-based counterparts, such as directly using the full task instructions (Full-Instr.) and semantic-role labeling (SRL).
> We even additionally compare our full framework with the ground truth OUC phrases obtained from both datasets, where our framework still performs generally better.
>
> [4] Plummer, Bryan A., Matthew Brown, and Svetlana Lazebnik. "Enhancing video summarization via vision-language embedding." *CVPR* 2017
>
> [5] Hessel, Jack, et al. "A case study on combining asr and visual features for generating instructional video captions." *CoNLL* 2019
>
> [6] Chen and Lin et al, “Joint Multimedia Event Extraction from Video and Article” *EMNLP* 2021
>
> [7] Peng, Xiaokang, et al. "Balanced multimodal learning via on-the-fly gradient modulation." *CVPR* 2022
>
> [8] Li, Liunian Harold, et al. "Grounded language-image pre-training." *CVPR* 2022
>
> [9] Zhang, Haotian, et al. "Glipv2: Unifying localization and vision-language understanding." *NeurIPS* 2022
>
> ---
>
> **[W3] Details of TREK-150**
>
> Thank you for bringing up this point, we would like to take this opportunity to clarify that TREK-150 is an **“object tracking”** dataset adapted to our setting (adaptation details in Sec 4.2 L436-L455), where given an instructed action, the model is required to ground and track the OUC throughout all frames in the egocentric video (L166-177). Therefore, unlike the Ego4D SCOD dataset, it **does not contain any defined Pre/PNR/Post frames**. Since our model is trained to perform joint inference and autonomously decide which of the pre- and post-conditions to weigh more based on the frame image and instructed action, **frame-type information is not required for our inference**.
>
> We will be sure to include this clarification in the final version of our paper. Also please feel free to let us know if you have any further questions, we will be happy to answer in detail and add them to the final version of our paper! We believe these revisions can be easily performed in the camera-ready version and hope you may kindly re-evaluate this point as a reason to reject our work.

---

### Official Review · Reviewer_MyHn · 2023-08-11

**Soundness:** 4

**Excitement:**

4: Strong: This paper deepens the understanding of some phenomenon or lowers the barriers to an existing research direction.

**Paper Topic And Main Contributions:**

The paper approaches the "active target grounding task", which involves identifying and grounding the object undergoing change in a sequence of images or video. The authors propose a two stage approach, in using ChatGPT to first identify object mentions from the instructions of a video, and then grounding it using GLIP.

**Reasons To Accept:**

- The paper is written well and easy to follow.
- The results and analysis are very detailed, and the chosen baselines for comparison seem appropriate.
- The authors provide a good amount of detail on limitations of their approach, opening up multiple areas for future study.

**Reasons To Reject:**

Some comments -

- Visual failure cases when the OUC does not to get detected would be nice to see.
- Cases when the OUC goes out of the egocentric view don't seem to be considered. For instance, if the object being spoken about in the instruction moves out of the video for a while and comes back, it is not clear what the GLIP output would be like.
- While the ablation on GPT-prompts is interesting, the results for all of them seem close to each other. The inference seems unnecessary.

**Reproducibility:**

4: Could mostly reproduce the results, but there may be some variation because of sample variance or minor variations in their interpretation of the protocol or method.

**Reviewer Confidence:**

3: Pretty sure, but there's a chance I missed something. Although I have a good feel for this area in general, I did not carefully check the paper's details, e.g., the math, experimental design, or novelty.

---

> ### Author Rebuttal · Authors · 2023-08-29
>
> We are thankful to Reviewer MyHn for the insightful remarks! We are encouraged that you find **our paper well-written and easy to follow**, **our baselines appropriate**, and **our analysis detailed** while **opening up multiple areas for future research**!
>
> Please find your suggestions and concerns addressed below:
>
> ---
>
> **[W1] Visual failure cases**
>
> We absolutely agree with the reviewer on the importance of analyzing visual failure cases, which is precisely why we provide a **comprehensive set of qualitative examples** in Figure 6 under Appendix B.1 (not in the main paper due to space constraints).
> In Figure 6 (C) and (F), we observe failure cases where our method correctly localizes part of the OUC but not its whole, and hence fails to align perfectly with the ground truth bounding boxes.
> This type of failure can be mostly attributed to the object detector having difficulty in identifying the OUCs’ differently colored/structured regions.
>
> Other failure cases presented in Figure 6 mostly occur when GPT extracted symbolic knowledge is suboptimal (as analyzed in Table 1), or when drastic shifts in object views (drastically shifted from the objects’ normal views) or lighting conditions of objects occur.
> Given more space for our final version, we will include detailed qualitative analysis of each failure case in the main paper.
>
> ---
>
> **[W2] OUC exiting egocentric view**
>
> We are happy that the reviewer brings up the issue of object occlusion, which is something we have also observed and carefully examined during our experiments!
> Due to the space limitations, we list our key findings below for your reference and will add them as further details in our final version:
>
> - **Ego4d SCOD**: Pre/PNR/Post frames have considerably low likelihood of exhibiting object occlusion. However, if the object to be grounded is out of the PoV, the GLIP predicted object logits would almost certainly be **lower than a pre-set objectness threshold** (due to the learning process), and hence lead to a ***“no-prediction”***.
>
> - **TREK-150 (Epic-Kitchens)** videos do exhibit the phenomenon of the OUC going “in-and-out” of the egocentric PoV, resulting in *partial occlusion* and/or *full occlusion* frames where no ground truth annotations for the OUC are provided.
> Such frames are excluded from the final evaluation.
> Our model is very successful in predicting the objects when they come back due to the robustness of our symbolic joint inference grounding mechanism.
>
> ---
>
>
> **[W3] Ablation on GPT-prompts**
>
> Thanks to the reviewer for bringing up this question! We would like to first provide some context for better understanding the performance gap between our GPT-only model and the GPT+symbolic knowledge model: In terms of AP50, the performance gain for OUC in TREK-150 is 2.45 across all frames, the corresponding gain for OUC and Tool in Ego4D SCOD across all frames are 1.53~1.74 and 5.71～5.83, respectively. Judging from the fact that the results are obtained from ~38K video frames for Ego4D SCOD and ~90K frames from TREK-150, the performance gain on AP-style metrics should be quite significant for the OUC, and even more so for Tool.
>
> Furthermore, our method is not limited to the existing datasets/works, where we aim at devising a generally applicable framework that future works can utilize, perhaps on more challenging active object grounding datasets, where the objects potentially go through more drastic state changes.

---

### Meta-Review · Area_Chair_yDLz · 2023-09-19

**Recommendation:** 4

**Metareview:**

The paper proposes a method to use world knowledge from LLMs to improve active object detection and tracking in ego-centric videos.

Strengths:
- Paper is well written (MyHn)
- Interesting idea (h4o5)
- Results and analysis (MyHn, 5jwp)
- Detailed limitations section (MyHn)
- Generality of method (h4o5)

The concerns raised by the reviewers were mostly comments to provide constructive feedback, or mostly addressed during rebuttal (reviewers h4o5, 5jwp). No serious outstanding soundness concerns are identified and there is overall consensus on the soundness of the paper.

---

### Decision · Program_Chairs · 2023-10-07

**Decision:**

Accept-Main

**Comment:**

The paper proposes a method to use world knowledge from LLMs to improve active object detection and tracking in ego-centric videos.

Strengths:
- Paper is well written (MyHn)
- Interesting idea (h4o5)
- Results and analysis (MyHn, 5jwp)
- Detailed limitations section (MyHn)
- Generality of method (h4o5)

The concerns raised by the reviewers were mostly comments to provide constructive feedback, or mostly addressed during rebuttal (reviewers h4o5, 5jwp). No serious outstanding soundness concerns are identified and there is overall consensus on the soundness of the paper.